# Testing Biodegradable Films as Alternatives to Plastic-Film Mulching for Enhancing the Yield and Economic Benefits of Processed Tomato in Xinjiang Region

**Anwar Abduwaiti** [1]**, Xiaowei Liu** [2]**, Changrong Yan** [3,4]**, Yinghao Xue** [5]**, Tuo Jin** [5]**, Hongqi Wu** [2]**, Pengcheng He** [6]**, Zhe Bao** [5,*] **and Qin Liu** [3,4,*]

1 Xinjiang Uygur Autonomous Region Agricultural Resources and Environmental Protection Station, Urumqi 830049, China; 18690323206@163.com
2 College of Grass and Environmental Sciences, Xinjiang Agricultural University, Urumqi 830052, China; lxw1012@126.com (X.L.); hqwu7475@126.com (H.W.)
3 Institute of Environment and Sustainable Development in Agriculture, Chinese Academy of Agricultural Sciences, Beijing 100081, China; yanchangrong@caas.cn
4 Key Laboratory of Prevention and Control of Residual Pollution in Agricultural Film, Ministry of Agriculture and Rural Affairs, Beijing 100081, China
5 Rural Energy and Environment Agency, Ministry of Agriculture and Rural Affairs, Beijing 100125, China; xueyinghao@agri.gov.cn (Y.X.); jintuo273@126.com (T.J.)
6 Inner Mongolia Autonomous Region Agricultural Technology Extension Station of Ulanqab, Ulanqab 012000, China; nybstzzhbc@163.com
* Correspondence: baozhe1983@126.com (Z.B.); liuqin02@caas.cn (Q.L.); Tel./Fax: +86-10-82109773 (Q.L.)

**Abstract:** The extensive application of plastic-film mulching (PFM) has brought a series of environmental pollution due to the lack of awareness of plastic-film rational use and absence of plastic residues recycling in China. In addition, the use of degradable film instead of common polyethylene plastic film (PE film) can effectively alleviate this situation. The substitution of PE film with biodegradable film in the agricultural production of processed tomato in Xinjiang region was investigated in this study. Using bare soil as the control, we compared the effects of PE film and biodegradable film mulching on crop growth, yield, and economic benefits in processed tomato. The results indicated that: (1) Biodegradable film with a thickness of about 8 μm can meet the mechanical operation requirements, and the effect of biodegradable film mulching was completely consistent with that of PE film; (2) Four kinds of biodegradable film can meet the requirements of processed tomato growth and development, although slightly different from PE film in increasing temperature and water retention; (3) Plastic-film planting can ensure a net profit of 1.14–1.64 ten thousand CNY per hectare under the current production conditions and mode of Xinjiang region, and biodegradable film planting was observed to be essentially equal to those of PE film; (4) Nearly 50%–70% of the biodegradable film was ruptured and degraded during processed tomato harvesting, which avoided the occurrence of the winch of the plastic-film winding harvester and improves the efficiency and commodity rate of the processed tomato harvest operation. As the biodegradable film mulching causes no residual pollution, it is accepted to be an alternative to plastic-film mulching for agricultural applications and supports the sustainable development of agroecosystems in Xinjiang region.

**Keywords:** biodegradable film; degradation rate; yield; economic benefit; processed tomato; Xinjiang region

## 1. Introduction

Plastic-film mulching (PFM) is widely accepted to be one of the important technologies used to promote the significant improvement of agricultural productivity and great change of production system, with the functions of increasing topsoil temperature, decreasing soil evaporation, inhibiting the growth of weeds and salt accumulation, promoting earlier

germination, increasing illumination uniformity of crops canopy, and scattered light [1–5]. PFM was promoted for use on cotton in Xinjiang region of China in the early 1980s. In recent years, PFM technology has been further developed with the advancement of drip-irrigation technology and supporting agricultural machinery [6]. In 2019, the plastic-film-covered area in the Xinjiang region was 0.35 million ha, and this resulted in a plastic-film usage of 0.24 million tons, which accounted for 17.6% of national plastic-film use. As a result, Xinjiang region has become the province with both the highest plastic-film use and largest cultivation area in China [2,7,8].

Meanwhile, the extensive application of PFM has resulted in widespread environmental pollution due to a lack of awareness related to rational use of plastic film and the absence of plastic residue recycling [9–11]. Furthermore, most of the plastic film remains directly in the soil or is burned after use. There is no doubt that these practices release harmful substances and have a negative impact on the environment. The main component of the film used in agricultural production in China is linear low-density polyethylene (LLDPE) or low-density polyethylene (LDPE), whose molecular structure is very stable [12–14]. Moreover, it is easy to break during application in the field because of the thinness of the plastic film, which results in both low secondary utilization of plastic film and accumulation of the residual film in the soil [15–17]. The accumulation of large amounts of residual film leads to changes in the soil physical structure, hinders the downward transport of soil water and nutrients, and decreases soil porosity and permeability, resulting in a decline in the quality of cultivated land [16–19]; furthermore, the accumulated residual film increases the risk of farmland microplastic pollution [20–23]. The effect of biodegradable film is similar to that of common plastic film (PE film) in increasing temperature and preserving soil moisture, considering the fact that PFM cannot be replaced and the lack of effective means of recycling residual film [24]. The development and application of environment-friendly new products of biodegradable film is an important technical way to realize clean agricultural production and ensure food security in the future.

Biodegradable film has been widely accepted to be a kind of plastic film degraded by microorganisms in the natural environment. Japanese scientists defined biodegradable film as "polymers and their blends that can be decomposed into low molecular compounds, that will not have adverse effects on the environment, through the action of microorganisms in nature" [25]. The main components of biodegradable films based on petroleum include dicarboxylic acid diol copolyester (PBAT, etc.) [25–27], polyhydroxyalkanoate (PHA) [28], polycaprolactone (PCL) [29], polyhydroxybutyrate (PHB) [30,31], and carbon dioxide copolymer—polypropylene carbonate (PPC) [32]. These polymers can be quickly decomposed and used by microorganisms in nature, and the final degradation products are carbon dioxide and water [24,33]. At present, biodegradable film has been used in the production of greenhouse crops [34,35], potato [36–38], spring peanut [39], cotton [40–42], maize [42–45], rice [46,47], and winter wheat [48,49]. As a result, it has been found that biodegradable film can reduce soil residual film pollution, as it is able to fulfill the main functions of PFM: heat preservation, moisture conservation, crop growth, and yield promotion [15,50]. However, there are differences in the degradation characteristics and the increase or decrease of crop yield after degradable film mulching, which may be influenced by different crop types, film mulching methods, irrigation amounts, and regional climatic conditions [51]. The major aims of this study were as follows: (1) investigate whether the positive impacts of biodegradable mulch film in Xinjiang region on crop growth and yield could match those of common plastic film; (2) comparatively analyze the effect of biodegradable mulch film and common plastic film on economic benefits; (3) and comparatively analyze the degradation of biodegradable mulch film and common plastic film.

## 2. Test Materials and Methods

### 2.1. General Situation of the Research Area

The experiment area is located in Changji City, Xinjiang Uygur Autonomous region, as shown in Figure 1, in the middle of the northern slope of Tianshan Mountain and on the southern margin of Junggar Basin. The experimental site was Xiabahu Farm in Changji City, which is located in the westerly belt of middle latitudes. It has typical continental climate characteristics, such as cold and long winters, hot summers, low precipitation, strong evaporation, and annual and diurnal range of air temperature. In addition, the annual average temperature is 6.8 °C, the extreme maximum temperature is 42.0 °C, and the extreme minimum temperature is −38.2 °C. Moreover, the annual average precipitation is 180.1 mm, and the evaporation is 2390 mm.

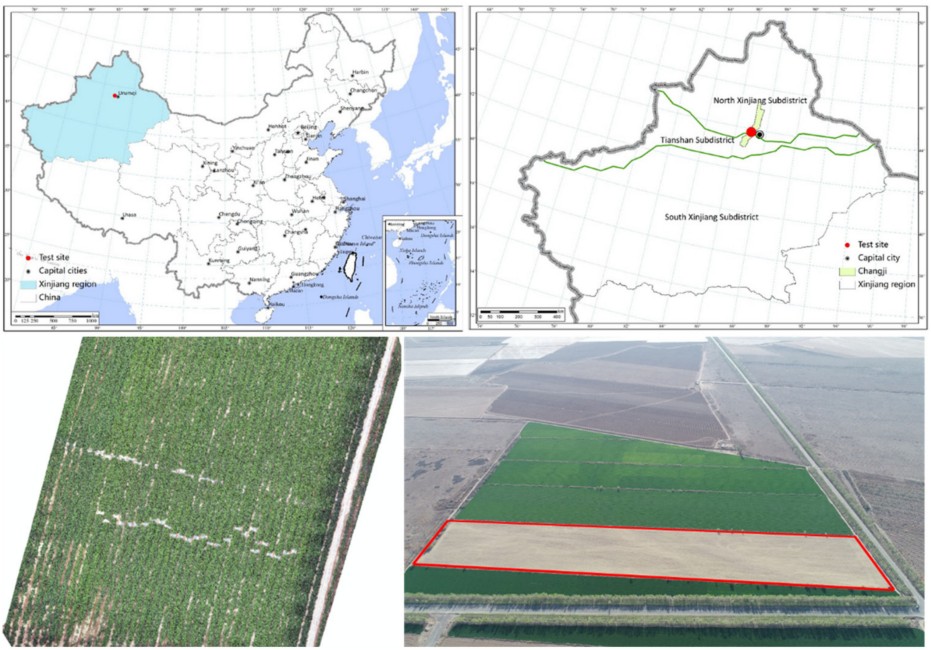

**Figure 1.** Location of the experiment site.

The experimental area was 3.0 ha (Figure 1). The local soil was medium loam with a pH of 7.67. In the surface soil (0–20 cm depth), total N, P, and available K were 0.81 g·kg$^{-1}$, 0.76 g·kg$^{-1}$, and 0.32 g·kg$^{-1}$, respectively. The site has been practicing mulched drip irrigation since 2013. The average soil bulk density of the topsoil was 1.15 g·cm$^{-3}$.

### 2.2. Materials and Methods

#### 2.2.1. Test Materials

The main raw materials of biodegradable mulch film were poly (adipic acid)/butylene terephthalate (PBAT). Biodegradable films from four manufacturers, Baishan Xifeng (BF1), Shandong Qingtian (BF2), Shanghai Hongrui (BF3), and Lanshan Tunhe (BF4), were tested in the present study. The physical parameters of biodegradable mulch films are shown in Table 1.

**Table 1.** The information of biodegradable mulch film.

| Mulch Film Types | Color | Thickness (μm) | Width (cm) | Maximum Load (N, Vertical/Horizontal) |
|---|---|---|---|---|
| Baishan Xifeng (BF1) | Black | 9.6 | 125 | 4.1/1.7 (>1.5) |
| Shandong Qingtian (BF2) | Black | 9.8 | 125 | 4.2/1.6 (>1.5) |
| Shanghai Hongrui (BF3) | Black | 9.8 | 125 | 3.2/1.6 (>1.5) |
| Lanshan Tunhe (BF4) | Black | 9.7 | 125 | 4.2/1.7 (>1.5) |
| PE | Black | 10 | 125 | - (1.6) |

2.2.2. Experimental Design

Four kinds of biodegradable films, PE film, and open field cultivation were utilized, comprising a total of six treatments. Each treatment comprised an experimental plot with three repeats, randomly arranged using single factor random block design. Each treatment plot was 30 m in length and 20 m in width. In addition, one tomato seedling was transplanted in each hole with nutrient matrix, and drip irrigation was carried out before artificial transplantation.

During the whole growth period of processed tomato, nine cycles of drip irrigation and fertilization were carried out every 7 days from June 1 to August 3, 2019–2020. Weeds were removed, the soil was loosened at the beginning of June, and drip irrigation was applied under the plastic film with the integration of water and fertilizer. Moreover, the amount of fertilizer application was N 300–375 kg/ha, $P_2O_5$ 300–375 kg/ha, and $K_2O$ 375 kg/ha.

*2.3. Observation Index and Method*

2.3.1. Strength Performance of Plastic Film

The conventional film mulching machine was used to test whether the plastic film met the operational requirements of agricultural machinery according to the actual situation of local plastic-film laying. In addition, the investigation and evaluation of the film under the normal walking state of the film mulching machine were undertaken to assess whether there was fracture, adhesion, and various other important parameters.

2.3.2. Temperature Increasing Performance

The temperature automatic recording probe device was used to measure the temperature (accuracy 0.2 °C) according to the requirements for the determination of ground temperature in the Code for Surface Meteorological observation (ISBN:9787502936907) of the National Meteorological Administration of China. The buried depth of the probe was 10 cm, and the data were recorded once every 60 min, with three repetitions. Finally, the daily average temperature was calculated.

2.3.3. Soil Moisture Conservation Performance

In addition, the moisture conservation performance of plastic film was tested by water-vapor-transmittance test method [52], using a PERME W3/060 water-vapor-transmittance tester with a test accuracy of 0.01 g/(m²·24 h). Analysis of each sample was repeated three times, each repeated six times, and its stable value was the moisture permeability of the plastic film.

2.3.4. Observation on Degradation of Biodegradable Film

The damage to the biodegradable film was observed every 10 days and recorded (presence of weather cracks, cracks, the degree of fragmentation, the number of cracks, and the number of broken pieces). In addition, the degradation was determined, focusing on recording the occurrence date of the induction period in the first 30 days after film mulching.

Moreover, the scenes of each degradation stage were photographed and recorded as follows: there were three fixed observation points for each treatment, and the fixed frame was used for fixed-point photography (50 cm × 50 cm). Three fixed observation points in a representative repetition were selected to take pictures until the next crop was prepared, and these long-range photos were taken at an observation interval of 5 m as per the requirement.

The degradation degree of biodegradable film on the ridge (border) surface of each treatment plot was observed, including the overall condition, residual film strength test, and evaluation during crop harvest.

### 2.3.5. Determination of Yield and Quality of Processed Tomato

In addition, the growth period of the pilot processed tomato was investigated, and the date of commencement of the growth period was defined as the day when more than 50% of the plants in the plot entered the growth period. Three points, each of 10 m$^2$, were selected for each experimental treatment, and six treatments were repeated for three times with a total of 18 points. The yield per hectare was calculated after the yield was measured. For the determination of the number of tomatoes per plant: five consecutive processed tomato plants were selected in the sampling area to determine the number of single bead fruits. The weight of a single tomato fruit was calculated by averaging the total weight of collected processed tomato fruits. In each sample, moderately mature tomatoes were selected to determine the main quality parameters, including vitamin C (VC) content, total acid, sugar–acid ratio, and total sugar content. Moreover, the contents of VC, total acid, and total sugar in processed tomato were determined by UV, neutralization, and sulfuric acid-anthrone methods, respectively.

### 2.4. Data Analysis

The single-factor variance and multiple comparative analysis to determine statistical differences between the effects of different mulching films on the yield and economic benefits of processed tomato were performed using SPSS19.0 software. Differences were considered statistically significant when $p \leq 0.05$.

## 3. Results and Analysis

### 3.1. Machine Performance of Biodegradable Film

Four kinds of biodegradable films and one kind of PE plastic film were assessed for use in the routine operation of local processed tomato planting. The operation was carried out by using a 1MK-3 plastic-film covering and punching machine, each operation covering three ridges (125 cm). The processed tomato was planted manually. The results showed that the four kinds of tested biodegradable films had no problems such as fractures and adhesion in the process of mechanical mulching, and the operation was smooth, which was basically consistent with PE plastic film. In addition, the results showed that biodegradable film could fully meet the requirements of mechanical mulching in the production of processed tomato in Xinjiang, China.

### 3.2. Effect of Biodegradable Film on Ground Temperature

As described in Figure 2, the effect of plastic-film mulching on farmland soil temperature is not simple. However, it was closely related to climatic conditions and tomato growth period. Generally, the soil temperature of plastic-film mulching was higher than that of open-field cultivation in the early stage (before late June). However, the temperature of open-field cultivation was higher than that of plastic-film-mulched farmland after late June, which was mainly because there was no ridge sealing for processed tomato planting in an open field. The sun can shine directly on the soil, resulting in higher soil temperature. By contrast, the soil temperature was lower in the processed tomato farmland covered with plastic film, due to the dense growth of processed tomato and the interception of solar radiation by the canopy.

Plastic-film mulching increased the soil accumulated temperature of processed tomato by 63.2–116.7 °C in the 36 days before June 10. In addition, plastic-film mulching also reduced the soil accumulated temperature by 3.1–50.0 °C during the monitoring period from June 10 to late July. This fully shows that the effect of plastic film cannot be measured simply by warming. In addition, Figure 1 also demonstrated that the warming performance of biodegradable film was weaker than that of PE film. Meanwhile, the warming performance of different biodegradable films was also different, with the warming property of Shandong Qingtian biodegradable film being relatively weak.

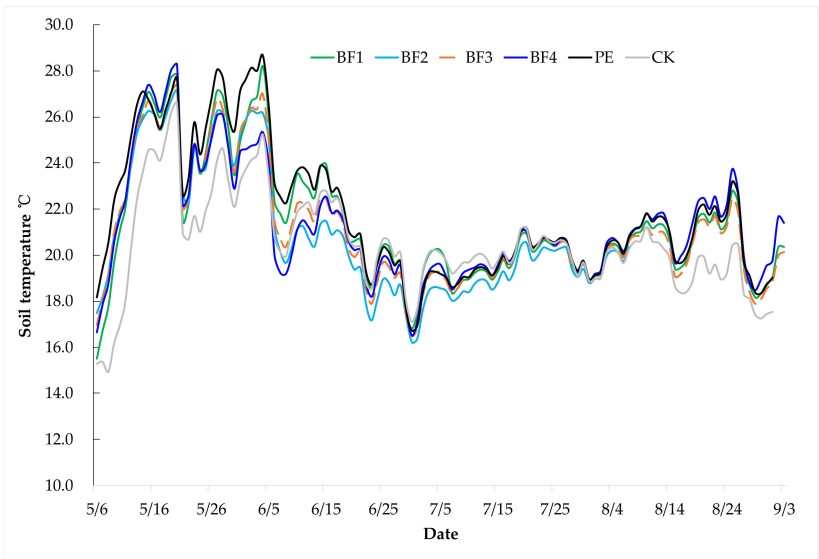

**Figure 2.** Temporal characteristics in soil temperature of processed tomato under different mulching films.

### 3.3. Soil Moisture Conservation Characteristics of Biodegradable Film

Moisture retention is recognized to be another important function of plastic-film mulching. How to reduce soil water loss after plastic-film mulching is an important aspect to consider when improving the formula of biodegradable film due to the raw material differences compared with PE film. In Figure 3, the moisture permeability data showed that the water content reduction rate of biodegradable film was significantly higher than that of PE film. The moisture permeability of biodegradable film of BF1, BF3, and BF4 was about 190 g/(m$^2$·24 h) when the biodegradable film was kept intact, whereas that of BF2 was about 210 g/(m$^2$·24 h). According to the national standard of biodegradable film GB/T35795–2017, the moisture permeability of biodegradable film with a thickness of less than 10 μm is less than 800 g/(m$^2$·24 h). In this experiment, the moisture permeability of biodegradable film was much lower than this standard. It indicates that the water-retention performance of biodegradable film in China has been greatly improved. In addition, there was no difference in the water retention of different biodegradable films in the production of processed tomato because the technology of drip irrigation under plastic film was used, which better ensures the water supply needed for growth and development. However, attention should be paid to this challenge of meeting adequate water supply in dry farming areas.

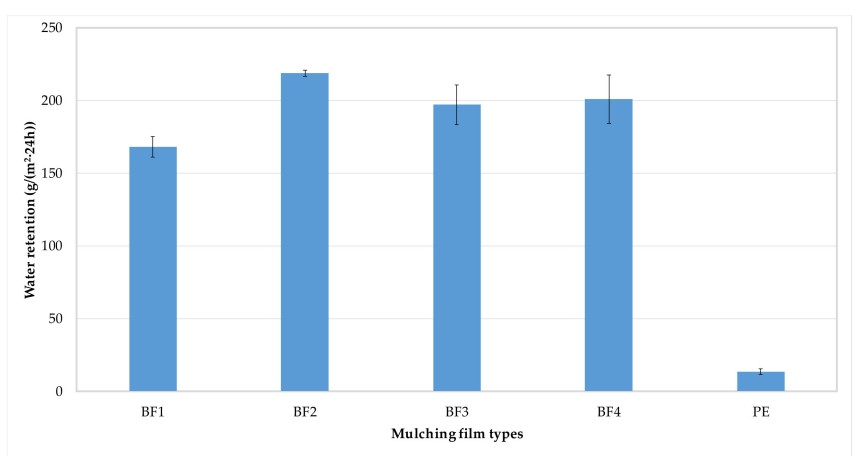

**Figure 3.** Characteristics of water retention for different biodegradable mulching films.

### 3.4. Degradation of Biodegradable Film

The fixed-point observation results showed that the Lanshan Tunhe biodegradable film began to develop small pores in late June and degraded 65 days after mulching. Larger pores appeared at the end of June, but as a whole, the biodegradable film remained relatively intact. A large area of rupture and degradation occurred after late July, and some part buried in soil was basically degraded in the first 10 days of October after harvest. However, the mechanical strength was effectively lost, although the film was exposed to the surface.

Shanghai Hongrui biodegradable film began to develop small holes in early June and started to degrade about 50 days after mulching, and a large-area rupture and degradation appeared at the end of June. Large areas of film were effectively degraded after late September compared with Lanshan Tunhe biodegradable film. The degradation rate of Shandong Qingtian, Baishan Xifeng, and Lanshan Tunhe biodegradable films were basically the same, with a large area of rupture and degradation occurring in the middle and last 10 days of July and a large area of film degradation occurring after the last 10 days of September. Generally, the degradation process of biodegradable films followed the same pattern: the emergence of small pores at first, then larger pores and large-area rupture, and finally, they broke into small pieces. The degradation process was accompanied by thinning, brittleness, and loss of mechanical strength in the film. Shandong Qingtian and Lanshan Tunhe biodegradable films began to degrade about 65 days after film mulching, whereas Shanghai Hongrui biodegradable film began to degrade 10 days ahead of schedule. At the same time, the degradation area reached 60% during the harvest period of processed tomato. Moreover, some part buried in soil was basically degraded in the first 10 days of October after harvest.

### 3.5. Effects of Film Mulching on Yield and Quality of Processed Tomato

The growth period of processed tomato covered with biodegradable film and PE film was basically the same, and there was no significant difference between treatments. However, the ripening period of processed tomato covered with Shanghai Hongrui biodegradable film was slightly delayed, which may be related to its premature rupture, degradation, and the decrease of water retention. As a result, it revealed that (except for yield) Shanghai Hongrui biodegradable film was significantly worse than other biodegradable films for mulching processed tomato. The yield level of processed tomato mulched with other biodegradable films was not significantly different from that with PE film mulching, and the yield of all plastic-film mulching treatments was higher than that of open-field planting. The use of plastic-film mulching seriously affected the yield of processed tomato per plant as compared to open-field cultivation (Table 2).

**Table 2.** Yield of processed tomato under different biodegradable film mulching.

| Treatments | Fruit Number per Hill | Fruit Weight per Hill (kg) | Single Fruit Weight (g) | Yield (t/ha) |
|---|---|---|---|---|
| BF1 | 62.2 b | 3.54 ab | 56.3 a | 125.7 a |
| BF2 | 64.8 ab | 3.72 a | 57.1 a | 132.0 a |
| BF3 | 61.7 b | 3.38 b | 55.1 ab | 119.8 b |
| BF4 | 66.7 a | 3.62 a | 54.3 ab | 127.5 a |
| PE | 67.1 a | 3.39 b | 50.5 b | 124.1 a |
| CK | 55.8 c | 2.91 c | 52.2 b | 104.3 c |

Notes: Different letters in the same column indicate significant differences ($p < 0.05$).

As depicted in Table 3, the VC content of processed tomato planted in an open field was higher than those in plastic-film mulching. There was little difference between different plastic films. The sugar content and acidity reflect the flavor quality of tomato; the results showed that plastic-film mulching could increase the sugar content and sugar–acid ratio of tomato. BF2 and PE film mulching produced the sweetest taste, and the sugar–acid

ratio reached 13.8, whereas in planting in an open field, it was only 10.2, and the total sugar content also decreased significantly. It indicated that plastic-film mulching had a significant effect on the product quality of processed tomato.

**Table 3.** Quality of processed tomato under different biodegradable film mulching.

| Treatments | Total Acid (%) | VC (%) | Sugar–Acid Ratio (%) | Sugar–Acid Ratio |
|---|---|---|---|---|
| BF1 | 0.40 ab | 24.32 b | 5.08 a | 12.7 |
| BF2 | 0.39 b | 26.65 ab | 5.37 a | 13.8 |
| BF3 | 0.36 b | 26.11 ab | 5.16 a | 12.9 |
| BF4 | 0.37 b | 22.23 b | 4.79 a | 12.8 |
| PE | 0.41 ab | 25.75 b | 5.68 a | 13.8 |
| CK | 0.46 a | 32.11 a | 4.69 a | 10.2 |

Notes: Different letters in the same column indicate significant differences ($p < 0.05$).

### 3.6. Economic Feasibility Analysis of Biodegradable Film

The production inputs of processed tomato in Xinjiang mainly include substances (such as fertilizer, plastic film, seedlings, pesticides, and irrigation water), secondary labor force, land rent, fruit harvest and transportation, and plastic-film recovery. The input per hectare ranges from 3.43 ten thousand CNY to 3.54 ten thousand CNY, of which plastic-film investment accounts for 2.9%–7.1% of the total input costs, a relatively low proportion. In addition, the proportion is closely related to the type and dosage of plastic film. For example, according to the need for plastic film of 0.83 ten thousand $m^2$ per ha, the unit price of biodegradable film was 25 CNY/kg and PE film 12 CNY/kg, and the input cost of plastic film per hectare was 1000–2500 CNY (Table 4). Moreover, the information of input–output for processed tomato was obtained under biodegradable films mulching (Table 5).

**Table 4.** Information of producer goods and labor cost for processed tomato in Xinjiang region (ten thousand CNY/ha).

| Treatments | Agricultural Material Inputs | | | | | Leasehold and Labor Cost | | Harvesting and Plastic Recycling Cost | | Total Inputs |
|---|---|---|---|---|---|---|---|---|---|---|
| | Fertilizer | Plastic Film | Germchit | Pesticides | Water and Electricity | Labor Cost | Leasehold Cost | Harvesting Cost | Plastic Recycling Cost | |
| BF1 | 0.36 | 0.25 | 0.48 | 0.08 | 0.27 | 0.45 | 0.60 | 1.05 | | 3.54 a |
| BF2 | 0.36 | 0.25 | 0.48 | 0.08 | 0.27 | 0.45 | 0.60 | 1.05 | | 3.54 a |
| BF3 | 0.36 | 0.25 | 0.48 | 0.08 | 0.27 | 0.45 | 0.60 | 1.05 | | 3.54 a |
| BF4 | 0.36 | 0.25 | 0.48 | 0.08 | 0.27 | 0.45 | 0.60 | 1.05 | | 3.54 a |
| PE | 0.36 | 0.10 | 0.48 | 0.08 | 0.27 | 0.45 | 0.60 | 1.05 | 0.05 | 3.43 a |
| CK | 0.36 | 0.00 | 0.48 | 0.08 | 0.33 | 0.60 | 0.60 | 1.05 | | 3.50 a |

Notes: The difference in land rent was relatively large, generally between 0.6–0.9 ten thousand CNY per ha, and the film back charge was 0.05–0.08 ten thousand CNY per ha. Different letters in the same column indicate significant differences ($p < 0.05$).

**Table 5.** Information of input–output for processed tomato under different biodegradable film mulching (ten thousand CNY/ha).

| Treatments | Total Inputs | Yield (t/hm²) | Total Output | Profit | Profit Rate (%) |
|---|---|---|---|---|---|
| BF1 | 3.54 a | 125.7 a | 4.96 a | 1.42 b | 12.8 |
| BF2 | 3.54 a | 132.0 a | 5.18 a | 1.64 a | 30.2 |
| BF3 | 3.54 a | 119.8 b | 4.67 b | 1.14 c | −9.7 |
| BF4 | 3.54 a | 127.5 a | 4.97 a | 1.41 b | 14.1 |
| PE | 3.43 a | 124.1 a | 4.84 a | 1.26 c | 11.9 |
| CK | 3.50 a | 104.3 c | 4.07 c | 0.57 d | −54.5 |

Notes: Different letters in the same column indicate significant differences ($p < 0.05$).

## 4. Discussion

Plastic-film mulching was widely accepted to be a cultivation technique to improve crop-growth environment through artificial method and can obviously increase temperature and preserve soil moisture. It can improve soil water and heat, nutrient-use efficiency, and soil fertility. Finally, it can achieve the effect of stable yield and efficiency [2,53]. The cumulative pollution of residual film has a negative impact on the soil and ecological environment in Xinjiang, especially in northern Xinjiang. The effect of biodegradable films on heat preservation and water retention was similar to that of PE film, the yield level of biodegradable film was similar to that of PE film, and they also had the ability to naturally degrade [24,43,54].

The degradation of biodegradable film made from the same material but produced by different manufacturers is slightly different. For example, this study revealed that Shanghai Hongrui biodegradable film began to degrade about 50 days after film mulching, whereas Shandong Qingtian and Lanshan Tunhe biodegradable mulching films began to degrade about 65 days after film mulching, and the degradation area could reach 60% during the harvest period of processed tomato. In addition, Shen et al. [43] studied a kind of photo-biodegradable film, which showed that cracks began to appear at 30–40 days after mulching, and a large area of cracking occurred after 90 days, and the thin film degraded fast. Moreover, Zhang et al. [40] reported that the cotton biodegradable mulch film with polylactic acid as raw material entered the induction period after 17–22 days of mulching, gradually entered the breaking stage after 60 days, and entered the collapse stage after about 130 days. The induction period of biodegradable film using PBAT as raw material was longer than that of the photo-biodegradable film and polylactic acid (PLA). In this research, biodegradable film with an induction period of 65 days can meet the planting demand of processed tomato under transplanting conditions in Xinjiang.

In addition, soil temperature is an important factor that directly or indirectly affects the growth, development, and yield of crops. The soil temperature at 0–10 cm beneath the soil surface plays an important role in the survival of processed tomato seedlings in early spring. The daily average soil temperature before the rupture of the biodegradable film was 1.23 °C lower than that of the common PE film but 2.26 °C higher than that of the open field, respectively. There was no ridge sealing, so the temperature of open soil was higher in the later stage of sun exposure due to the lower temperature in the early stage of processed tomato development in open land. Generally, the soil thermal-insulation performance of biodegradable film was better before cracking, which was equivalent to that of PE film, consistent with the conclusions of Liu [55] and Shen [43]. Consistently with our results, Caruso et al. [34] and Ibarra et al. [56] found a positive correlation between the crop earliness and the soil heat accumulation under mulching treatments and no significant differences between the plastic and biodegradable film. As a consequence, plastic-film mulching can increase the effective accumulated temperature of the soil [54,57].

The yield of processed tomato covered by biodegradable film from different manufacturers was different compared with PE film. The yield with biodegradable film mulching increased by 6.0% in Shandong Qingtian and 2.4% in Lanshan Tunhe. There was no significant difference in the yield of processed tomato between the two manufacturers. Shanghai Hongrui biodegradable film decreased processed-tomato production by 13.3%. Liu et al. [55] found that biodegradable film mulching could increase maize yield by 18.7%, which was slightly higher than that of PE film mulching (17.7%). He et al. [58] found that cotton yield with biodegradable film mulching in Shihezi, Xinjiang was 2.8% higher than that of PE film mulching. In addition, Duan et al. [36] also found that the tuber yield of potato covered with two kinds of biodegradable films increased by more than 20%, which was much higher than 6.3% of PE film mulching. The reason for this difference may be due to different types of biodegradable film materials, crop types, and regional environmental conditions. This suggests that the biodegradable mulch is a good environmentally friendly alternative to the plastic one for managing processed tomato in Xinjiang region; however, all the mulching treatments encouraged the production compared to bare soil, which

highlights the importance of both increasing soil temperature and controlling weeds with no costly manual intervention [34,35,59].

As one of the important ways to solve the pollution of residual plastic film, the application of biodegradable film has excellent effects and great potential in agricultural production. However, there are still technical problems. At present, it is necessary to strengthen the research on the raw materials, formula, and production technology of biodegradable film to improve product quality and reduce cost, especially to develop special biodegradable film products for specific regions and crops [24]. It intends to meet and adapt to the requirements of agricultural production diversity. In this research, Lanshan Tunhe and Shandong Qingtian biofilms were economically feasible in Xinjiang processed-tomato production and have market prospects for promotion under the current production conditions and mode of Changji, Xinjiang.

## 5. Conclusions

1.  Biodegradable mulch film with a thickness of about 8 μm can meet the mechanical operation requirements, and the effect of biodegradable mulching film was completely consistent with that of PE film. There were no problems such as fracture and tearing of biodegradable film in the process of operation.
2.  In addition, although the four kinds of biodegradable mulch film in this experiment were slightly different from PE film in increasing temperature and water retention, they basically meet the requirements of processed-tomato growth and development. Nearly 50%–70% of the biodegradable film was ruptured and degraded during processed-tomato harvesting, which avoids the occurrence of the winch of the plastic-film winding harvester and improves the efficiency and commodity rate of the processed tomato harvest operation.
3.  Plastic-film planting can ensure a net profit of 1.14–1.64 ten thousand CNY per hectare under the current production conditions and mode of Changji in Xinjiang. In addition, the yield, output value, and net profit of processed tomato covered with three kinds of BF1, BF2, and BF4 were the same as those of PE film. This indicated that the application of biodegradable film in the production of processed tomato in Xinjiang is economical and feasible.

**Author Contributions:** Z.B., Q.L., X.L., C.Y., and Y.X. mostly contributed equally at designing of experiment. A.A., T.J., H.W., and P.H. carried out data collection and processing. A.A., Z.B., and Q.L. compiled and edited the full text this paper. All authors have read and agreed to the published version of the manuscript.

**Funding:** This research was supported by the Zero-Waste Agricultural Mulch Films for Crops in China project (Newton Fund: Newton UK-China Agritech Challenge 2017; No. 2017YFE0121900), NERC GCRF Plastics proposal call: Reducing the Impacts of Plastic Waste in Developing Countries (Do agricultural microplastics undermine food security and sustainable development in less economically developed countries? NE/V005871/1) and Central Public-Interest Scientific Institution Basal Research Fund (Y2019LM02-06). We gratefully acknowledge the anonymous reviewers for their valuable comments on the manuscript.

**Institutional Review Board Statement:** Not applicable.

**Informed Consent Statement:** Not applicable.

**Data Availability Statement:** Not applicable.

**Acknowledgments:** We gratefully acknowledge the anonymous reviewers for their valuable comments on the manuscript.

**Conflicts of Interest:** The authors declare no conflict of interest.

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
