# Peer review of "Testing Biodegradable Films as Alternatives to Plastic-Film Mulching for Enhancing the Yield and Economic Benefits of Processed Tomato in Xinjiang Region"

_sustainability, doi:10.3390/su13063093_

Round 1

Reviewer 1 Report

Dear Authors, you should address my comments highlighted inside the 
manuscript.

Author Response

Dear Editor,

Thank you for your letter of decision and for the referee’s comments concerning our manuscript entitled “Testing Biodegradable Films as Alternatives to Plastic Film Mulching in Enhancing the Yield and Economic Benefits of Processed Tomato in Xinjiang Region” (NO: sustainability-1122706). We have studied the comments carefully and have corrected our manuscript accordingly. Please find below an overview of the changes with respect to the comments.

Dear Authors, you should address my comments highlighted inside the manuscript.

Reviewer 1. We revised our manuscript according to the reviewer’s suggestion highlighted inside the manuscript. In an addtion, we read carefully the recommended literatures. We added “Consistently with our results, Caruso et al. [34] and Ibarra et al. [56] found a positive cor-relation between the crop earliness and the soil heat accumulation under mulching treat-ments and no significant differences between the plastic and biodegradable film.” and “This suggests that the biodegradable mulch is a good environmentally friendly alternative to the plastic one for managing processed tomato in Xinjiang region; however, all the mulching treatments encouraged the production compared to bare soil, which highlights the importance of both increasing soil temperature and controlling weeds with no costly manual intervention [34, 35, 59].” in the discussion.

We revised 2.4 Data Analysis as “The sigle-factor variance and multiple comparative analysis to determine statistical differences between the effects of different mulching films on the yield and econmic bene-fits of processed tomato were performed using SPSS19.0 software. Differences were con-sidered statistically significant when p≤0.05.”

We have been made successfully the following modifications:

  1. We have successfully edited grammatical revision;
  2. We added notes for Table 2, Table 3 and Table 5.

Additionally, details in the paper have been improved according to the reviewer's suggestion.

Once again, thank you very much for your contribution on our manuscript.

Yours sincerely,

Associate Prof. Qin Liu

Reviewer 2 Report

Dear editor and dear colleagues,
I have now concluded my review on the submitted manuscript “Testing Biodegradable Films as Alternatives to Plastic Mulching Film in Enhancing the Yield and Economic Benefits of Processed Tomato in Xinjiang Region”

It is a review study that focuses on the study  of the extensive application of plastic mulching film (PMF) has brought a series of environmental pollution due to the lack of awareness of plastic film rational use and absence of plastic residues recycling in China, and the use of degradable film instead of common polyethylene plastic film (PE film) can effectively alleviate this situation.

According to my opinion, this is a well-organized manuscript, addressing a very interesting and original work –it has merit for publication. Moreover, it is well structured, focused and well written.

I have a few observations:

- It would be desirable to describe the type of soil, the paper indicates that it is a "bare soil", and it would have been interesting to carry out studies with different types of soils, as well as extending the study to two years, and not just one agricultural season.

- Also in section 2.3.4. it indicates "The damage to the plastic film was observed every 10 days and recorded", the data found should have been better reported, indicating for example % of deterioration, since it is one of the objectives foreseen in the study: “The damage to the plastic film was observed every 10 days and recorded".

- Once the test has been set up, it could be used to determine other parameters, such as chlorophyll content and canopy temperatura.

- In the stated objectives it indicates: "; (2) comparatively analyze the effect of biodegradable mulch film and common plastic film on economic benefits", therefore the information related to the economic impact should be expanded.

Based on the above reasons my recommendation is a minor revisión.

Author Response

Dear Editor,

Thank you for your letter of decision and for the referee’s comments concerning our manuscript entitled “Testing Biodegradable Films as Alternatives to Plastic Film Mulching in Enhancing the Yield and Economic Benefits of Processed Tomato in Xinjiang Region” (NO: sustainability-1122706). We have studied the comments carefully and have corrected our manuscript accordingly. Please find below an overview of the changes with respect to the comments.

Reviewer 1. I have now concluded my review on the submitted manuscript “Testing Biodegradable Films as Alternatives to Plastic Mulching Film in Enhancing the Yield and Economic Benefits of ProcessedTomato in Xinjiang Region”.

It is a review study that focuses on the study of the extensive application of plastic mulching film(PMF) has brought a series of environmental pollution due to the lack of awareness of plastic filmrational use and absence of plastic residues recycling in China, and the use of degradable filminstead of common polyethylene plastic film (PE film) can effectively alleviate this situation.

According to my opinion, this is a well-organized manuscript, addressing a very interesting andoriginal work –it has merit for publication. Moreover, it is well structured, focused and well written.

I have a few observations:

Comment 1: It would be desirable to describe the type of soil, the paper indicates that it is a "bare soil", and it would have been interesting to carry out studies with different types of soils, as well as extending the study to two years, and not just one agricultural season.

Response: We added “The experimental area was 3.0 ha (Figure 1). The local soil was medium loam with a pH of 7.67. In the surface soil (0–20 cm depth), total N, P, and available K was 0.81 g·kg−1, 0.76 g·kg−1, and 0.32 g·kg−1, respectively. The site has been practiced mulched drip irrigation since 2013. The average soil bulk density of the topsoil was 1.15 g·cm−3.” in section “2.1. General Situation of the Research Area”.

Comment 2: Also in section 2.3.4. it indicates "The damage to the plastic film was observed every 10 days and recorded", the data found should have been better reported, indicating for example % of deterioration, since it is one of the objectives foreseen in the study: “The damage to the plasticfilm was observed every 10 days and recorded"

Response: In our manuscript, the damage to the biodegradable film was observed every 10 days and recorded (presence of weather cracks, cracks, the degree of fragmentation, the number of cracks, and the number of broken pieces). Moreover, as described in follwing Figure as an example, the scenes of each degradation stage were photographed and recorded as follows: there were three fixed observation points for each treatment, and the fixed frame was used for fixed-point photography (50 cm × 50 cm). We found that the degradation of Lanshan Tunhe biodegradable film was characterized in a large area of rupture and degradation occurring in the middle and last 10 days of July and a large area of film degradation occurring after the last 10 days of September.

The advice of “indicating for example % of deterioration of biodegradable film” from the referee is very good. Indeed, we tried to take the samples of biodegradable films (50 cm × 50 cm) in sync with the fixed-point photography and transported to the laboratory for being weighed after being washed in the water. Finally, we failed to get the expected fluctuation in weight loss rate of biodegradable films, because the adhered soil, fertilizer and pesticides can’t be washed off from the biodegradable film samples.

We discussed about this issue in our team group. In our experiment plan of 2021, weight loss of biodegradable films will be obtained through calculating the thickness plus the density plus the area of collected samples in sync with the photography every 10 days after film mulching.

Figure Degradation characteristics for biodegradable mulching film of Lanshan Tunhe (BF4). Notes: the name of biodegradable mulching film was marked by Chinese language as “蓝山屯河”.

Comment 3: Once the test has been set up, it could be used to determine other parameters, such as chlorophyll content and canopy temperatura.

Response: The referee’s advice is very good. The test of chlorophyll content, canopy temperatura, leaf area index of processed tomato and irrigation will be included in our experiment schedule of 2021 to get more supporting information for major aims of our study.

Comment 4: In the stated objectives it indicates: "; (2) comparatively analyze the effect of biodegradable mulch film and common plastic film on economic benefits", therefore the information related to the economic impact should be expanded.

Response: “Agricultural material inputs”, “Leasehold and labor cost”, and “Harvesting and plastic recycling cost” were selected for “Total inputs” in our manuscript considering the availability of influence factors. More factors such as irrigation, the subentry of labor cost will be included in coming research.

We have been made successfully the following modifications:

  1. We have successfully edited grammatical revision;
  2. We have revised the references.

Additionally, details in the paper have been improved according to the reviewer's suggestion.

Once again, thank you very much for your contribution on our manuscript.

Yours sincerely,

Associate Prof. Qin Liu

Round 2

Reviewer 1 Report

Dear Authors, you have addressed my comments and your manuscript can be accepted for publication.